# Machine Learning Improves the Prediction of Responses to Immune Checkpoint Inhibitors in Metastatic Melanoma

**DOI:** 10.3390/cancers15102700

**Published:** 2023-05-10

**Authors:** Azadeh Tabari, Meredith Cox, Brian D’Amore, Arian Mansur, Harika Dabbara, Genevieve Boland, Michael S. Gee, Dania Daye

**Affiliations:** 1Department of Radiology, Massachusetts General Hospital, 55 Fruit Street, Boston, MA 02114, USA; 2Harvard Medical School, Boston, MA 02215, USA; 3Department of Surgery, Massachusetts General Hospital, 55 Fruit Street, Boston, MA 02114, USA; 4Boston University Chobanian & Avedisian School of Medicine, 72 East Concord Street, Boston, MA 02118, USA

**Keywords:** melanoma, LDH, machine learning, combined models, immunotherapy

## Abstract

**Simple Summary:**

Lactate dehydrogenase (LDH) levels prior to treatment are a known biomarker to predict advanced melanoma’s response to immune checkpoint inhibitors (ICI). In this study, we evaluated the ability of machine learning-based models to predict responses to ICI and complement LDH for in predicting the outcomes of metastatic melanoma. A machine learning algorithm was developed using radiomics, and further analysis helped select the most important predictive features and variables. The machine learning model that combined both features extracted from images (radiomics) and pretreatment LDH levels resulted in better predictions of cancer response to ICI than models that only use radiomics features or LDH levels alone.

**Abstract:**

Pretreatment LDH is a standard prognostic biomarker for advanced melanoma and is associated with response to ICI. We assessed the role of machine learning-based radiomics in predicting responses to ICI and in complementing LDH for prognostication of metastatic melanoma. From 2008–2022, 79 patients with 168 metastatic hepatic lesions were identified. All patients had arterial phase CT images 1-month prior to initiation of ICI. Response to ICI was assessed on follow-up CT at 3 months using RECIST criteria. A machine learning algorithm was developed using radiomics. Maximum relevance minimum redundancy (mRMR) was used to select features. ROC analysis and logistic regression analyses evaluated performance. Shapley additive explanations were used to identify the variables that are the most important in predicting a response. mRMR selection revealed 15 features that are associated with a response to ICI. The machine learning model combining both radiomics features and pretreatment LDH resulted in better performance for response prediction compared to models that included radiomics or LDH alone (AUC of 0.89 (95% CI: [0.76–0.99]) vs. 0.81 (95% CI: [0.65–0.94]) and 0.81 (95% CI: [0.72–0.91]), respectively). Using SHAP analysis, LDH and two GLSZM were the most predictive of the outcome. Pre-treatment CT radiomic features performed equally well to serum LDH in predicting treatment response.

## 1. Introduction

The incidence of metastatic melanoma, which has a 5-year survival rate of 32%, is estimated to double every decade [1,2]. The survival benefit of targeted therapies as well as immunotherapy, including ipilimumab (CTLA-4 checkpoint inhibitor), nivolumab and pembrolizumab (PD-1 checkpoint inhibitors), has been proven [3]. The main limitations of the use of immune checkpoint inhibitors (ICI) in these patients include individual variations in treatment response and survival improvement, and the absence of objective, accurate predictive biomarkers of response [1,2,3].

Serum lactate dehydrogenase (LDH) is the main biomarker of response to immunotherapies; however [4,5,6], an increase of LDH serum level cannot differentiate pseudoprogression (defined as transient size increase followed by tumor response) from true progression. Therefore, therapeutic changes cannot be always initiated based on the changes in LDH serum level [7,8]. Identifying reliable predictive biomarkers for better monitoring of patients undergoing ICI therapy and identifying the patients likely to benefit from ICI therapy would be a major advance in pairing patients with beneficial therapies [9,10,11,12,13]. A recent study showed that melanoma tumors that have high imaging heterogeneity exhibited varied treatment responses and decreased survival [14,15]. Currently, changes in therapy are made according to imaging size-based response assessment, RECIST 1.1 (response evaluation criteria in solid tumors), and on follow-up CT examinations [16,17]. Interpretations of most malignant tumors with heterogenous growth and response patterns remains an issue of clinical decision, especially because of the limitations of RECIST 1.1 [18,19,20,21]. Intra-tumoral heterogeneity and tumor infiltration are newer imaging features that can be spatially assessed on imaging using quantitative measures of image signal heterogeneity and may improve prognostication compared with size assessment alone [22,23]. In routine practice iRECIST criteria, defined according to immune-related response criteria, have been introduced to partially overcome the heterogenous response problem and assess response to ICI therapy on contrast-enhanced CT scans, although these criteria have not been validated and atypical responses have been observed [23,24].

Several studies have evaluated the prognostic value of blood and imaging biomarkers, but the prognostic potential of combined imaging and blood markers has not been investigated in metastatic melanoma [11,25,26,27,28]. We hypothesized that baseline CT-based quantitative radiomics solely and combined with serum LDH can be predictive of responses to immunotherapy. In this study, we assessed the role of machine learning-based radiomics in predicting responses to ICI and in complementing LDH for prognostication of metastatic melanoma.

## 2. Materials and Methods

This study was approved by the Massachusetts General Brigham Institutional Review Board (IRB), and was conducted in accordance with the Health Insurance Portability and Accountability Act (HIPAA) guidelines for research. The requirement for patient informed consent was waived by the Massachusetts General Brigham IRB. The study was performed in accordance with the Declaration of Helsinki.

### 2.1. Patients and Clinical Data

Abdominal CT studies over a 14-year period (January 2008–January 2022) were reviewed in a retrospective manner for the presence of secondary malignant hepatic tumors from metastatic melanoma in the 1-month period prior to initiation of ICI. Clinical data collected at follow-up included medical and surgical interventions, and disease response to treatment (clinical outcomes). Inclusion criteria consisted of (a) age greater ≥18 years, (b) histopathology-confirmed melanoma cancer, (c) presence of biopsy-proven liver metastases on the arterial phase images of abdomen CT performed within 1 months prior to starting treatment, (d) available pre-immunotherapy serum LDH level, and (e) patients treated by an oncologist in our institution’s Cancer Center to ensure the availability of a response to immunotherapy and follow-up data. The presence of a response to immunotherapy was defined using RECIST version 1.1. At 3 months post-treatment evaluation, treatment response was defined by stable disease, and partial or complete response. According to RECIST criteria, the absence of a treatment response was defined by progression. Patients were then designated to 1 of the 2 groups: (1) Responders and (2) Non-responders [17]. 

Seventy-nine eligible patients (F: M 28:51; mean age 62 ± 17 years; range, 27–90 years) with 168 liver metastases were identified. None of the patients received treatment (surgery, immunotherapy, or targeted therapies) for their liver metastases before CT examinations. 

Clinicopathological data included patient age, patient gender, pre-immunotherapy LDH level, tumor NRAS and BRAF mutation status, primary site of disease, tumor staging, RECIST data, and ICI type received, and this data were obtained from the electronic clinical and histological records by the study coordinator who was blinded to the radiological data. 

### 2.2. CT Acquisition

All patients underwent a contrast enhanced abdominal exam on a 64-row (Discovery HD 750; GE Healthcare, Chicago, IL, USA) or 128-row (Somatom Definition Flash, Siemens Healthineer, Erlangen, Germany) multidetector CT scanner. A volume of 2 mL/kg of non-ionic contrast material (iomeprol, 350 mg iodine/mL; Iomeron 350, Bracco) was injected into an antecubital vein at a flow rate of 3.5 mL/s, followed by 50 mL of saline solution at the same flow rate. Acquisition parameters were as follows: tube voltage: 120 and 100 kVp; automatic tube current modulation: 64 × 1.25 mm and 128 × 1.25 mm; slice thickness: 5 mm.

### 2.3. Segmentation and Image Processing

Images were reviewed on the Picture Achieving and Communication System (PACS) by a radiologist, blinded to all clinical information, to identify metastatic lesions >1 cm in greatest diameter. From one to nine target metastases for each patient were selected following RECIST criteria. A volumetric image analysis software (3D slicer; http://www.slicer.org, accessed on 14 October 2022) was used to perform manual delineations and volume-based hepatic lesion segmentation on DICOM images [29] by a diagnostic radiology instructor (AT, 6 years of experience in image segmentation), on the original arterial phase contrast-enhanced CT images. An experienced fellowship-trained radiologist reviewed the segmentations (DD, 8 years of experience). 

### 2.4. Extraction of Radiomic Features and Machine Learning Model Training

The 116 radiomic features, including first-order, 2D and 3D shape features, gray level co-occurrence matrix (GLCM) features, gray level size zone matrix (GLSZM), gray level run time length matrix (GLRLM) features, neighboring gray tone difference matrix (NGTDM) features, and gray level dependence matrix features (GLDM), were extracted from 168 hepatic metastatic lesions [30,31]. The image analysis pipeline is summarized in Figure 1. 

A random forest machine learning algorithm using a selected subset of all radiomics features was used to predict the tumor response to immunotherapy. Maximum relevance minimum redundancy (mRMR) was used to select features that are both correlated with the outcome and non-redundant with other features in the set. Features with over 90% missing values were removed. The dataset was divided into a training, validation, and testing set at a ratio of 3:1:1. To avoid bias from excluding images with missing feature values, optimal imputation [32] was used to impute missing values. Feature sets of sizes 1 to the total number of features were selected with mRMR on the training data, and for each feature set, a new random forest model was developed using training data and evaluated using validation data. To balance the data for model development, synthetic cases of the minority class were generated using ADASYN [33]. The model that exhibited the best performance utilized fifteen features. The square root of the number of features indicated the depth of the random forest algorithm. Area under the receiver operating characteristic (AUC) analysis, confusion matrix metrics, and Brier score were used to assess and compare classification performance. The Brier score measures the mean of the squared difference between the predicted score and the occurrence of response to ICI. A lower Brier score indicates a more accurate model [34]. Youden’s index was used to select an optimal threshold for generating a confusion matrix [35].

Logistic regression analysis was used for the LDH model. Additionally, a combined machine learning algorithm based on all radiomics features and pre-treatment LDH was used to predict the response to immunotherapy.

To understand which of the selected features most strongly influenced model predictions, Shapley additive explanations (SHAP) was applied with our model. SHAP is a game-theoretic approach to interpreting predictions of machine learning models and an extension of Shapley values, indicating the average marginal contribution of each feature over all combinations of features [36]. This allows for both local and global explanations of the model, indicating both feature importance and the direction of the relationship between a feature and model output. Python 3.8 (https://www.python.org/, accessed on 1 February 2023) was used to perform the statistical analyses and machine learning models.

## 3. Results

### 3.1. Patients and Tumor Characteristics

From 2008 to 2022, ICI were initiated in 303 patients with metastatic melanoma. Of these patients, 113 underwent contrast-enhanced abdomen CT within 3 months prior to start of the ICI. The final population who met our inclusion criteria and had hepatic metastatic disease was comprised 79 patients with 168 hepatic metastatic tumors. Figure 2 shows the study flowchart.

The patients’ characteristics are demonstrated in Table 1. Serum LDH level was elevated (>the normal upper limit i.e., 248 UI/L) in 36/79 patients (45.5%).

A total of 168 metastases were analyzed. The mean number of target metastases was 2.1 [1–9]. The average sum of lesions diameters was 24.5 [4.4–126.1] mm on pre-treatment CT. Disease progression was observed in 23/79 patients (29%) at three months after initiation of ICI.

The most common driver mutation in our cohort was BRAF, which was found in 23 patients (29%). Other relevant mutations included NRAS in 8 patients (10%). The majority of patients had cutaneous melanoma (72%), but several patients (28%) had other histologic subtypes of melanoma. 

### 3.2. ICI Response Kinetics for the Whole Population

Progression of disease was assessed for each tumor on follow-up CT at 3 months following treatment initiation using RECIST criteria. Of the 168 tumors, 100 (59.4%) were treated with ipilimumab, 30 (17.7%) with pembrozilumab, 25 (15%) with nivolumab, and 13 (2.8%) with a combination of ipilimumab and nivolumab as the first line treatment. All of the patients had an LDH assessment taken prior to start of ICI. The results showed that in non-responders, LDH levels were significantly higher when compared to the responding patients (*p* <0.01). This was in agreement with prior publications [18,19,23,24].

One hundred and twelve tumors (66.6%) responded to the ICI at 3 months follow-up (responding tumors) and 56 (33.3%) showed no response to ICI (progressing tumors). 

### 3.3. Comparison of Baseline Characteristics between Responding and Progressing Tumors

Of the 112 responding tumors, associated with 56 patients, 48 (42.6%) were treated with ipilimumab, 26 (23%) with pembrozilumab, 16 (14.6%) with nivolumab, and 22 (31.8%) with other ICI treatments. Fifty percent of patients (28/56) had multiple (>1) liver tumors. The volume (mean: 20.4 ± 13 [4.1–92] mL, *p* = 0.022) and maximum dimension (mean: 22 ± 14 [4.5–107] cm, *p* = 0.022) of the segmented tumors were significantly higher in responding tumors. The target lesion number ranged from 1 to 7. Although not significant, patients who failed to respond to ICI tended to be older than the responders (mean age 60 ± 15 [27–90] years vs. 63 ± 15 [22–89] years, *p* = 0.27). 

Of the 56 tumors associated with 23 patients that failed to respond to immunotherapy, 28 (50%) received ipilimumab, 13 (23.2%) nivolumab, 11 (19.6%) pembrozilumab, and 4 (7.2%) received other ICI treatments. The volume and maximum dimension of the segmented tumors was 37.3 ± 75 [4.4–562] mL and 29 ± 26 [4.5–126] in non-responding tumors, respectively. Eleven patients (48%) had >1 liver tumors. The target lesion number ranged from 1 to 9. Based on the tumor volume and number of liver lesions, the average overall liver disease burden was 0.013 ± 0.008 [0.002–0.06] in responders and 0.024 ± 0.05 [0.002–0.37] in non-responders.

Serum LDH levels were significantly lower in responders when compared to non-responders (mean 318 ± 214 [108–1242] UI/L vs. 914 ± 672 [160–2751] UI/L, *p* = 0.01). Fifty percent of responders were females, while most of those with progressing tumors (78%) were male (*p* = 0.01) (Table 2).

### 3.4. Predictive Power of Individual LDH, Radiomic and Combined Models

We included all radiomic features in the machine learning pipeline to predict treatment response in the study population. The dataset was divided into training (*n* = 100 tumors), validation (*n* = 34), and testing (*n* = 34 tumors) sets, with tumors from the same patients in the same set. Missing values were imputed separately in each set, and features were selected with mRMR. mRMR resulted in the selection of fifteen radiomics features, including 4 GLSZM and 4 GLDM, 3 GLCM, and 3 first-order and 1 shape, as shown in Figure 3A,B.

In the testing dataset, a random forest-based machine learning algorithm that included the above radiomic features yielded an AUC of 0.81 (95% CI: 0.65–0.94) (Figure 4A). Confusion matrix analysis revealed a sensitivity of 58%, a specificity of 91%, and an accuracy of 76%.

The logistic regression model based on the pretreatment LDH alone resulted in an AUC of 0.81 (95% CI: 0.72–0.91) in the testing dataset (Figure 4B) with a sensitivity of 70%, a specificity of 87%, and an accuracy of 76%. The machine learning model that included the combination of the selected radiomic features and the pretreatment LDH level achieved higher performance on all performance metrics including AUC, sensitivity, specificity, and accuracy measures, with an AUC of 0.89 (95% CI: 0.76–0.99), sensitivity of 75%, a specificity of 95%, and an accuracy of 85% in the testing dataset (Figure 4C) (Table 3).

The Brier score for the combined model was 0.121, lower than the models that include radiomics features alone (0.174) or LDH alone (0.155), indicating the higher accuracy of the model that combines both the radiomics features and LDH. Brier scores for the three models were obtained using the same independent test set. 

### 3.5. Radiomics-only and Combined Machine Learning Models Interpretation 

In the radiomics-only model, the most important feature as determined by SHAP was the Diagnostic Image Maximum, which is a first order statistic. For the combined model, in addition to LDH, the most important feature was GLSZM large area emphasis. This feature is a second order statistic that reports the quantity of homogeneous connected areas inside the volume, of a certain size and intensity, thus describing heterogeneity in the region of interest. A gray level zone is defined as the amount of connected voxels that share the same gray level intensity within the image region defined by the mask. The Large Area Emphasis feature calculates the maximum value in the normalized segmented image array. All features in order of importance, are pictured in the Shapley summary plot in Figure 5A,B, respectively.

## 4. Discussion

Our results illustrate that an explainable machine learning model that includes pre-treatment radiomic features and pre-treatment LDH provides better performance (AUC of 0.89) for predicting treatment responses to ICI. Additionally, our study has shed light on the independent predictive value of pre-treatment LDH and CT-based radiomics for early response to treatment according to RECIST criteria in patients with metastatic melanoma undergoing immunotherapy. 

Despite multiple previous translational and clinical research studies, there is still a critical need for more valid blood and imaging biomarkers for real-time assessments of responses to ICI, because once metastatic tumors have developed resistance to ICI by bypassing the host’s immune system control, the disease progression can be rapid.

For immunotherapy, prior studies have reported that elevated baseline LDH serum levels are correlated with poor prognosis and ICI response rates [5,37]. This is in agreement with our results: patients who responded to ICI had lower levels of LDH before initiation of the immunotherapy compared to non-responding patients (*p* = 0.00). However, there is evidence showing that the early response to ICI therapy and overall survival have not always been correlated with baseline serum LDH levels; this finding potentially broadens treatment accessibility to patients excluded on the basis of marker levels at baseline. Additional imaging biomarkers that can identify real-time changes in the disease status and predict the response to ICI for active monitoring are thus needed to allow for timely and personalized therapeutic adjustments. 

Radiomic features analysis can provide voxel-scale information about the compositional distribution profile within a tumor beyond anatomic size. In this context, several studies have highlighted the application of radiomics analysis to evaluate the efficacy and successful prediction of response to immunotherapy in variety of malignancies [38,39]. A recent study in patients with advanced NSCLC who were treated with ICI presented an approach that evaluates dynamic changes in specific tumor radiophenotypic attributes between baseline and post-treatment CT scans [40]. In addition, a few studies have extracted imaging biomarkers to assess the correlation between pretreatment CT texture parameters and survival prediction in patients with metastatic melanoma who received anti-PD-1 monoclonal antibodies [41,42,43,44]. Study findings by Trebeschi et al. on a small melanoma cohort proved an association between tumor textural radiomic patterns and responses to immunotherapy [45]. Durot et al. proposed that tumor skewness extracted from pre-treatment CT using texture analysis has been reported as a possible predictive biomarker of overall survival in metastatic melanoma patients who underwent anti-PD1 monoclonal antibody therapy [46].

In a study by Sun et al., an MRI-based radiomics model correlated with response to immunotherapy was developed and validated on 3 independent cohorts with diverse image acquisition parameters [47]. Both serum biomarkers and imaging findings are routinely investigated in patients with metastatic melanoma [48], but the combined predictive power of these markers is seldom investigated. To our knowledge, this is the first study that highlights the independent predictive value of combined LDH and machine learning-based radiomics in the assessment of response to ICI in these patients. Our analysis shows that such a multidisciplinary and multifaceted approach may have the potential to improve the treatment strategies and prognostic performance. 

This study has a number of limitations, including the relatively small number of patients and the lack of multi-institutional validation. Additionally, the reproducibility of texture analysis has been commonly problematic in the field of radiomics. The imaging data was normalized prior to analysis. Although separate training, testing, and validation sets were used, the study was trained, tested, and validated in a study population from the same institution. We seek to increase the diversity and volume of training and testing data and separately verify our model, prospectively and across multiple sites. 

In the next phase of this study, we will prospectively (1) review the CT images at two time points (before and 6–8 weeks after the start of ICI) to assess dynamic imaging biomarker change during treatment, (2) validate the results using an independent external dataset across multiple sites, and (3) reduce the inter-reader reliability by including >2 readers and automated segmentation to ensure the ability of producing the same results on a larger patient population, other scanners, and different image acquisition protocols.

## 5. Conclusions

This study identifies key radiomic features from a single pre-treatment CT scan that predict response to ICI in patients with metastatic melanoma using a machine learning-based model. The radiomics-only model performed with similar predictability as the model based solely on pretreatment serum LDH levels. Importantly, we showed that the combined machine learning-based model outperformed the models based on radiomic features and pretreatment serum LDH levels alone, and, therefore, has the potential to further optimize clinical practice through more precise and individualized treatment decisions. 

## Figures and Tables

**Figure 1 cancers-15-02700-f001:**
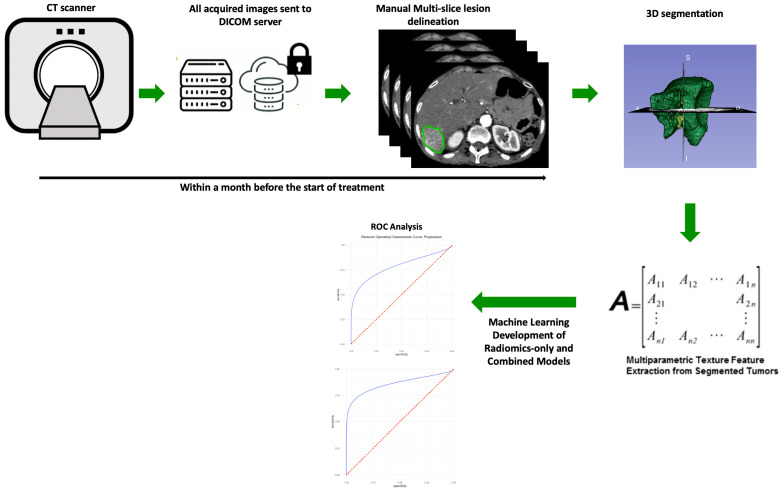
The image analysis pipeline used in this study consisted of tumor segmentation, feature extraction, and random forest-based machine learning for ICI therapy response prediction. Contrast-enhanced CT images in the arterial phase were used for analysis.

**Figure 2 cancers-15-02700-f002:**
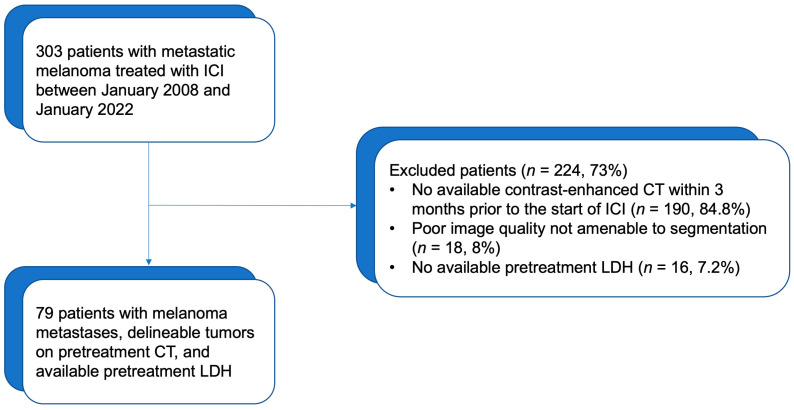
Flow diagram of data set.

**Figure 3 cancers-15-02700-f003:**
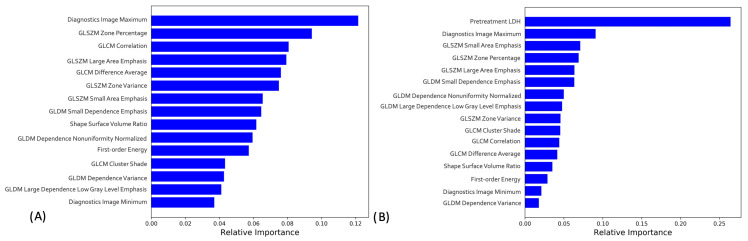
Top 15 radiomic features determined by mRMR for (**A**) radiomics only and (**B**) combined models.

**Figure 4 cancers-15-02700-f004:**
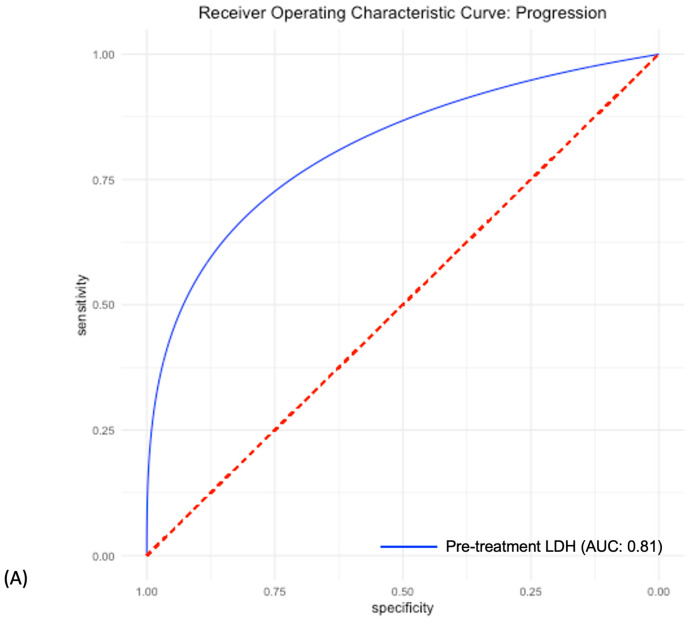
Receiver operating characteristic curves of the models based on the (**A**) Pre-treatment LDH, (**B**) Radiomic features, and (**C**) Pre-treatment LDH + radiomics feature.

**Figure 5 cancers-15-02700-f005:**
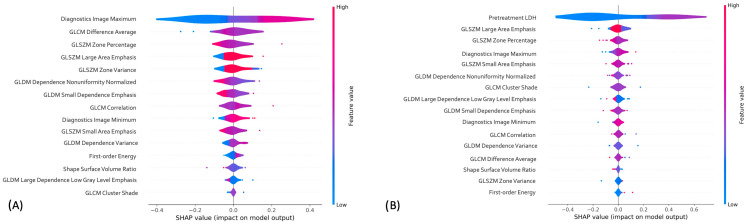
Shapley summary plot generated with Shapley additive explanation (SHAP) to illustrate the relative importance of each feature on (**A**) Radiomics-only and (**B**) combined model predictions.

**Table 1 cancers-15-02700-t001:** Patient, tumor, and treatment characteristics.

Variables	*n* (%)
Mean Age ^a^ (years ± SD)	62.1 ± 17
Gender	
Male	51 (64.6%)
Female	28 (35.4%)
NRAS mutation	
Positive	8 (10%)
Negative	71 (90%)
BRAF V600 mutation	
Positive	23 (29%)
Negative	56 (71%)
Primary Site of Disease	
Skin	57 (72%)
Other	22 (28%)
LDH > 248 (UI/L)	36 (45.5%)
Cancer Stage	4
Number of Liver Lesions	
1	41 (52%)
2	19 (24%)
3	9 (11.3%)
4	3 (3.7%)
5	3 (3.7%)
7	3 (3.7%)
9	1 (1.6%)
ICI Type	
Ipilimumab	47 (59.4%)
Pembrozilumab	14 (17.7%)
Nivolumab	12 (15%)
Other ICI	6 (7.9%)
RECIST status at three months	
Progression	23 (29%)
Stable, partial or complete response	56 (71%)

^a^ At the time of CT. ICI: immune checkpoint inhibitor; LDH: lactate dehydrogenase.

**Table 2 cancers-15-02700-t002:** Demographics features for the responders and non-responders.

Variable	Responders		Non-Responders		*p* Value
	Mean ± SD	Min–Max	Mean ± SD	Min–Max	
Age (years)	63 ± 15	22–89	60 ± 15	29–90	0.27
Gender (M:F)	28:28		18:5		0.01
LDH (UI/L)	318 ± 214	108–1242	914 ± 672	160–2751	0.00
Maximum dimension (cm)	22 ± 14	4.5–107	29 ± 26	4.5–126	0.027
Tumor volume (mL)	20.4 ± 13	4.1–92.7	37.3 ± 75	4.4–562	0.022

**Table 3 cancers-15-02700-t003:** Performance metrics of the three models.

Metric	LDH only	Radiomics Only	Combined
Sensitivity	0.70	0.58	0.75
Specificity	0.87	0.91	0.95
Accuracy	0.76	0.76	0.85
AUC	0.81 CI: [0.72–0.91]	0.81 CI: [0.65–0.94]	0.89 CI: [0.76–0.99]

## Data Availability

The data presented in this study are available on request from the corresponding author.

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
