# Peer review of "Machine Learning Improves the Prediction of Responses to Immune Checkpoint Inhibitors in Metastatic Melanoma"

_cancers, 2023, doi:10.3390/cancers15102700_

Round 1

Reviewer 1 Report

The manuscript is very interesting and has an approach that is inn line with the international discussion in the matter. 

Maybe the author can find further  input in the introduction or discussion considering the following references:

1. Madonna G,  Masucci GV,  Capone M,  Mallardo D,  Grimaldi AM,  Simeone E,  Vanella V,  Festino L,  Palla M,  Scarpato L,  Tuffanelli M,  D'angelo G,  Villabona L,  Krakowski I,  Eriksson H,  Simao F,  Lewensohn R,  Ascierto PA. Clinical Categorization Algorithm (CLICAL) and Machine Learning Approach (SRF-CLICAL) to Predict Clinical Benefit to Immunotherapy in Metastatic Melanoma Patients: Real-World Evidence from the Istituto Nazionale Tumori IRCCS Fondazione Pascale, Napoli, Italy. CANCERS 2021 13;16 
2.. Lindberg S,  Onjukka E,  Wersall P,  Staff C,  Lewensohn R,  Masucci G,  Lindberg K. Predicting the benefit of stereotactic body radiotherapy of colorectal cancer metastases. CLINICAL AND TRANSLATIONAL RADIATION ONCOLOGY 2022 36; 91-98

Author Response

The manuscript is very interesting and has an approach that is in line with the international discussion in the matter.
Maybe the author can find further input in the introduction or discussion considering the following references:
1. Madonna G, Masucci GV, Capone M, Mallardo D, Grimaldi AM, Simeone E, Vanella V, Festino L, Palla
M, Scarpato L, Tuffanelli M, D'angelo G, Villabona L, Krakowski I, Eriksson H, Simao F, Lewensohn R, Ascierto PA.
Clinical Categorization Algorithm (CLICAL) and Machine Learning Approach (SRF-CLICAL) to Predict Clinical Benefit to
Immunotherapy in Metastatic Melanoma Patients: Real-World Evidence from the Istituto Nazionale Tumori IRCCS
Fondazione Pascale, Napoli, Italy. CANCERS 2021 13;16
2.. Lindberg S, Onjukka E, Wersall P, Staff C, Lewensohn R, Masucci G, Lindberg K. Predicting the benefit of
stereotactic body radiotherapy of colorectal cancer metastases. CLINICAL AND TRANSLATIONAL RADIATION
ONCOLOGY 2022 36; 91-98
Thank you for the suggestion. We have cited the paper. 

Reviewer 2 Report

1. Please use updated information about melanoma survival over 5 years.  This has changed with the use of checkpoint blockade.  Your reference is from 2014.  

2. The slicer was utilized to determine manual delineations.  Based on this work can you propose a semi-automatic method for this step?  

3. It would be useful to characterize the types of melanomas in the other group.  Can any conclusions or hypotheses be made based on the type of melanoma?  

4. Should compare the relative efficacy of your population based on age versus the published literature.  You may not have enough power to state anything in this regard.  This is a complex question that should also take into account BRAF status.  

5. Minor comment:  The ROC curves need to have the font size increased and higher resolution.  

6. What would be a multiparameter test that could be tested in future studies given that the radiomics model "performed with similar predictability as the model based solely on pretreatment serum levels" ?  From the paper, it seems as if the thought is to combine the two, but this is not explicitly stated.  

7. Noticed that there is no funding for this study?  Was this funded by the Cancer Institute.  Who provided the protected time for the radiologist etc.

Author Response

1. Please use updated information about melanoma survival over 5 years. This has changed with the use of checkpoint
blockade. Your reference is from 2014.
Done. Thank you.
2. The slicer was utilized to determine manual delineations. Based on this work can you propose a semi-automatic
method for this step?
Thank you for the question.
Semi-automatic segmentation in 3D Slicer involves using both manual and automated methods to segment structures of
interest in medical images.
Manual tumor delineation and automatic tumor delineation using 3D Slicer have their respective advantages and
disadvantages. The choice between them depends on the specific requirements of the study, the available resources, and
the expertise of the users.
For this study, we performed manual tumor delineation involving a radiologist and an expert in medical image analysis
and segmentation to draw the tumor boundaries on each slice of the imaging study. This process required a high level of
expertise, attention to detail, and time, but it allowed for the incorporation of the radiologist's subjective assessment of the
tumor's size, shape, and location. This method can also account for variations in the tumor's appearance that may not be
captured by automated algorithms. In cases where accuracy and precision are crucial, such as in treatment planning or
clinical trials, manual delineation is often preferred.
3. It would be useful to characterize the types of melanomas in the other group. Can any conclusions or hypotheses be
made based on the type of melanoma?
Thank you for the interesting question.
In this study our focus was on the skin melanoma which accounted for 72% of the patients. The primary site of the
disease was ocular/uveal and vaginal only in 28% of patients. Validating the results from rarer sites of origin and
confirming the generalizability is an active area of investigation.
4. Should compare the relative efficacy of your population based on age versus the published literature. You may not
have enough power to state anything in this regard. This is a complex question that should also take into account BRAF
status.
We agree with the reviewer. However, this is beyond the scope of our study at this preliminary stage. We would consider
the mutation status in the next phase of this study.
5. Minor comment: The ROC curves need to have the font size increased and higher resolution.
We agree and have uploaded the high-resolution figures separately.
6. What would be a multiparameter test that could be tested in future studies given that the radiomics model "performed
with similar predictability as the model based solely on pretreatment serum levels"? From the paper, it seems as if the
thought is to combine the two, but this is not explicitly stated.

Thank you. The thought is to combine the two. The model we developed combines the radiomics features and
pretreatment serum LDH levels, and this combined model outperforms the radiomics model and the model based solely
on pretreatment serum levels. We have clarified this further in the conclusion.
7. Noticed that there is no funding for this study? Was this funded by the Cancer Institute. Who provided the protected
time for the radiologist etc.
Thank you for pointing this out. This study was supported by the RSNA Seed grant in 2021. We have incorporated this
information into the paper

Reviewer 3 Report

The authors present a paper about "Machine Learning Improves the Prediction of Responses to Immune Checkpoint Inhibitors in Metastatic Melanoma".

The topic is interesting becasue it combines two important fileds of resaerch; radiomics and immunotherapy.

The authors have adequately described the flowchart of their methods however there are a few points that I would like them to answer as follows:

1) In figure 1 the authors present a "manual multi-slice lesion delineation": it would important to provide further details with regard to the expertise of this process which is of paramount importance. The authors should state the expertise of the doctor who perfomed it (years of experience in the field), if an independent review was taken into account in case of doubt, only one person permoring the task or more than one?

2) In table 1 the author provide the number of liver metastasis but what about the overall burden of liver disease? it would be important to add data about the volume of liver disease for each patient indipendently from the number of lesions (3 small lesions may have a smaller burden of disease compared to 1 massive lesion)

3) Please move the conclusions to the discussion scetion and provide a real conlcusion to your preliminary study, not an intent for the future 

Author Response

The authors present a paper about "Machine Learning Improves the Prediction of Responses to Immune Checkpoint
Inhibitors in Metastatic Melanoma".
The topic is interesting becasue it combines two important fileds of resaerch; radiomics and immunotherapy.
The authors have adequately described the flowchart of their methods however there are a few points that I would like
them to answer as follows:
1) In figure 1 the authors present a "manual multi-slice lesion delineation": it would important to provide further details with
regard to the expertise of this process which is of paramount importance. The authors should state the expertise of the
doctor who perfomed it (years of experience in the field), if an independent review was taken into account in case of
doubt, only one person permoring the task or more than one?
Thank you for your question. As stated in the Methods section of the manuscript: “Manual delineations and volume-based
hepatic lesion segmentation on DICOM images by a diagnostic radiology instructor (AT, 6 years of experience in image
segmentation), on the original arterial phase contrast-enhanced CT images. An experienced fellowship-trained radiologist
reviewed the segmentations (DD, 8 years of experience)”
2) In table 1 the author provide the number of liver metastasis but what about the overall burden of liver disease? it would
be important to add data about the volume of liver disease for each patient indipendently from the number of lesions (3
small lesions may have a smaller burden of disease compared to 1 massive lesion)
Thank you for pointing out this interesting topic. Typical liver volume is ~1500 mL
(https://pubs.rsna.org/doi/10.1148/radiol.2021212010). Based on the tumor volume and number of liver lesions the
average overall liver disease burden was 0.013  0.008 [0.002-0.06] in responders and 0.024  0.05 [0.002-0.37] in nonresponders. We have included this into the manuscript.
3) Please move the conclusions to the discussion scetion and provide a real conlcusion to your preliminary study, not an
intent for the future
Thank you for the great suggestion. We have moved that to the Discussion.

Reviewer 4 Report

Authors combined LDH and AI-classifier to predict ICI response in melanoma patients. I think this kind of trial interesting to me since this method do not need extra testing (e.g. genetic testing). However, I have several concerns as follows:

1) So many drop-out patients (73%) seemed very odd.

2) The type of ICI used (CTLA-4 and/or PD-1) should be take into account when evaluating response since the response to these drug should differ.

3)  Also, the metastatic sit e greatly affect response. 

4) LDH indicates the volume of the tumor. Radiologic study might just reflecting the volume of the tumor.

5) To begin with, we do not say to the patient "your LDH is high, so you cannot use ICI". What I want to say is what is the merit for us when we can predict the response of ICI? 

Author Response

Authors combined LDH and AI-classifier to predict ICI response in melanoma patients. I think this kind of trial interesting
to me since this method do not need extra testing (e.g. genetic testing). However, I have several concerns as follows:
1) So many drop-out patients (73%) seemed very odd.
We agree with the reviewer that the number of excluded cases was high. However, it was crucial for the design of the
study at the preliminary stage to keep the dataset homogenous and consistent. In addition, the duration of this study was
2008 to 2022 and for many older cases we either did not have contrast-enhanced CT within 3 months prior to the start of
immunotherapy in our PACS system or the quality of CT images were not acceptable to perform precise segmentation.
2) The type of ICI used (CTLA-4 and/or PD-1) should be take into account when evaluating response since the response
to these drug should differ.
Thank you for the comment. The type of ICI was included in the combined model, however after mRMR feature selection,
it was not selected as the top 15 features of importance. Therefore, it is not selected as the top important features in the
combined SHAP plot as well.

3) Also, the metastatic sit e greatly affect response.
We agree with the reviewer that metastatic site beyond liver would impact the response. However, to ensure uniformity
have only focused on the liver metastasis at the preliminary stage of this study.
4) LDH indicates the volume of the tumor. Radiologic study might just reflecting the volume of the tumor.
. tumor burden is known to correlate with the LDH levels
Thank you. As suggested by Reviewer 3, we have calculated added the data about overall liver disease burden into the
text. Based on the tumor volume and number of liver lesions the average overall liver disease burden was 0.013  0.008
[0.002-0.06] in responders and 0.024  0.05 [0.002-0.37] in non-responders.
5) To begin with, we do not say to the patient "your LDH is high, so you cannot use ICI". What I want to say is what is the
merit for us when we can predict the response of ICI?
We thank the reviewer for this comment. We agree with the reviewer that we can not discontinue or not initiate ICI therapy
just because LDH levels are high. We have extensively explained in the second paragraph of the Introduction that
therapeutic changes cannot be initiated based on the changes in LDH serum level alone.
It is important to note that our combined model is based on multiple features and the model performance depends on the
radiographic features+LDH together. This will allow for better decision making and treatment management for patients
with metastatic melanoma

Round 2

Reviewer 2 Report

There is a paper that states that the survival with new immune therapies is at least 40% at 5 years.  There are several groups that have studied this.  Recommend picking one of them.  

Reviewer 3 Report

I have no further comments